# Endothelial Dysfunction Drives CRTd Outcome at 1-Year Follow-Up: A Novel Role as Biomarker for miR-130a-5p

**DOI:** 10.3390/ijms24021510

**Published:** 2023-01-12

**Authors:** Celestino Sardu, Gaetano Santulli, Gianluigi Savarese, Maria Consiglia Trotta, Cosimo Sacra, Matteo Santamaria, Mario Volpicelli, Antonio Ruocco, Ciro Mauro, Giuseppe Signoriello, Lorenza Marfella, Michele D’Amico, Raffaele Marfella, Giuseppe Paolisso

**Affiliations:** 1Department of Advanced Medical and Surgical Sciences, University of Campania “Luigi Vanvitelli”, 80126 Naples, Italy; 2Department of Medicine, Division of Cardiology, Albert Einstein College of Medicine, New York, NY 10461, USA; 3Department of Advanced Biomedical Sciences, “Federico II” University, 80131 Naples, Italy; 4Department of Medicine, Division of Cardiology, Karolinska Institutet, Heart, Vascular and Neuro Theme, Karolinska University Hospital, 17177 Stockholm, Sweden; 5Department of Experimental Medicine, University of Campania “Luigi Vanvitelli”, 80126 Naples, Italy; 6Cardiovascular and Arrhythmias Department “Gemelli Molise”, 86100 Campobasso, Italy; 7Cardiovascular Diseases and Electrophysiology Unit, “S. Maria della Pietà Hospital”, 80035 Naples, Italy; 8Cardiovascular Diseases and Electrophysiology Unit, “Cardarelli Hospital”, 80131 Naples, Italy; 9“Mediterranea Cardiocentro”, 80122 Naples, Italy

**Keywords:** HFrEF, CRTd response, miRs, inflammation, cardiac remodeling, clinical outcomes

## Abstract

Endothelial dysfunction (ED) causes worse prognoses in heart failure (HF) patients treated with cardiac resynchronization therapy (CRTd). ED triggers the downregulation of microRNA-130 (miR-130a-5p), which targets endothelin-1 (ET-1). Thus, we evaluated ED and the response to CRTd by assessing miR-130a-5p and ET-1 serum levels. We designed a prospective multi-center study with a 1-year follow-up to evaluate ED, ET-1, and miR-130a-5p in CRTd patients with ED (ED-CRTd) vs. patients without ED (NED-CRTd). Clinical outcomes were CRTd response, HF hospitalization, cardiac death, and all-cause death. At 1-year follow-up, NED-CRTd (n = 541) vs. ED-CRTd (n = 326) patients showed better clinical statuses, lower serum values of B type natriuretic peptide (BNP: 266.25 ± 10.8 vs. 297.43 ± 16.22 pg/mL; *p* < 0.05) and ET-1 (4.57 ± 0.17 vs. 5.41 ± 0.24 pmol/L; *p* < 0.05), and higher values of miR-130a-5p (0.51 ± 0.029 vs. 0.41 ± 0.034 A.U; *p* < 0.05). Compared with NED-CRTd patients, ED-CRTd patients were less likely to be CRTd responders (189 (58%) vs. 380 (70.2%); *p* < 0.05) and had higher rates of HF hospitalization (115 (35.3%) vs. 154 (28.5%); *p* < 0.05) and cardiac deaths (30 (9.2%) vs. 21 (3.9%); *p* < 0.05). Higher miR-130a-5p levels (HR 1.490, CI 95% [1.014–2.188]) significantly predicted CRTd response; the presence of hypertension (HR 0.818, CI 95% [0.669–0.999]), and displaying higher levels of ET-1 (HR 0.859, CI 98% [0.839–0.979]), lymphocytes (HR 0.820, CI 95% [0.758–0.987]), LVEF (HR 0.876, CI 95% [0.760–0.992]), and ED (HR 0.751, CI 95% [0.624–0.905]) predicted CRTd non-response. Higher serum miR-130a-5p levels (HR 0.332, CI 95% [0.347–0.804]) and use of ARNI (HR 0.319, CI 95% [0.310–0.572]) predicted lower risk of HF hospitalization, whereas hypertension (HR 1.818, CI 95% [1.720–2.907]), higher BNP levels (HR 1.210, CI 95% [1.000–1.401]), and presence of ED (HR 1.905, CI 95% [1.238–2.241]) predicted a higher risk of HF hospitalization. Hence, serum miR-130a-5p could identify different stages of ED and independently predict CRTd response, therefore representing a novel prognostic HF biomarker.

## 1. Introduction

Cardiac resynchronization therapy with defibrillation (CRTd) improves prognoses in patients with heart failure (HF) [1]. Endothelial dysfunction (ED) plays a crucial role in heart failure (HF), [1] and has been associated with worsening HF, hospitalizations, and cardiac deaths, independent of the baseline characteristics of CRTd patients [1]. ED is characterized by impaired, flow-mediated vasodilation (FMD), which is influenced by endothelium-dependent mechanisms, and has been shown to be associated with response to CRTd [2]. In this setting, serum endothelin-1 (ET-1) has been established as one of the main regulators of the endothelium, and a reliable marker of ED in patients with HF [3]. ET-1 is a vasoconstrictor produced from the prepropeptide big ET-1 by endothelin-converting enzyme, regulating vascular resistance in response to flow change and inflammatory stress [4]. Indeed, the binding of ET-1 to two G-protein coupled receptors (ETA and ETB) regulates the release of nitric oxide (NO) and/or increases its degradation, which then modulates vasoconstriction, cell proliferation, inflammation, and fibrosis [5]. Selective (ETA) and dual (ETA + ETB) receptor antagonists can improve NO bioavailability and endothelial function, and reduce inflammation in pathological situations [5]. Patients with chronic HF vs. controls exhibited higher cellular and serum expression of tumor necrosis factor (TNF-α), interleukin (IL-6), and ET-1 levels; the stimulation of TNF-α and IL-6 production in these patients led to the overexpression of ET-1 [6]. However, over-inflammation could promote the expression of ET-1, which could worsen ED and participate in the pathogenesis of cardiac failure [6].

ET-1 is known to be a negative prognostic marker for patients treated with an implantable cardioverter–defibrillator [4]. On these grounds, we hypothesized that modulating the expression of ET-1, and subsequently reducing ED, would result in better clinical outcomes in CRTd patients.

Emerging evidence indicates that ET-1 is directly targeted by microRNA-130 (miR-130a-5p), and preclinical assays suggest that the downregulation of ET-1 induced by miR-130a-5p leads to beneficial effects on cardiac remodeling [7,8]. Thus, we sought to determine whether CRTd patients with ED (ED-CRTd) vs. those without ED (NED-CRTd) would differentially express ET-1, inflammatory markers, and miR-130a-5p. Indeed, differential expression of these biomarkers might impact clinical outcomes in CRTd patients. Thus, in ED-CRTd vs. NED-CRTd patients, we evaluated: (i) serum levels of inflammatory markers, ET-1, and miR-130a-5p at baseline and at 12-month follow-up and (ii) clinical outcomes in terms of rate of CRTd responders, HF hospitalizations, cardiac deaths, and all-cause deaths.

## 2. Results

The characteristics of the study cohorts, at baseline and at end of follow-up, are reported in Table 1.

At baseline, ED-CRTd (n = 590) vs. NED-CRTd (n = 277) patients displayed higher values of ET-1 (*p* < 0.05). These cohorts did not show any significant difference in terms of clinical parameters, risk factors, comorbidities, inflammatory biomarkers, echocardiographic parameters, and medications (Table 1).

At 1-year follow-up, ED-CRTd (n = 326) vs. NED-CRTd (n = 541) patients exhibited significant differences in NYHA class, QRS duration, serum BNP, and ET-1, 6MWT, and serum miR130 levels (*p* < 0.05, Table 1). Specifically, miR-130a-5p levels were significantly increased, showing a 1.9-fold increase in NED-CRTd vs. a 1.3-fold increase in ED-CRTd (*p* < 0.05). Additionally, ED-CRTd patients had significantly higher values of lymphocytes and neutrophiles (inflammatory cells), CRP, IL-6, and TNF-α (inflammatory markers). ED-CRTd patients also had significantly lower LVEF and higher degrees of mitral insufficiency and other echocardiographic parameters (Table 1). Notably, investigating the differences (delta values, Δ) between the 1-year follow-up and baseline values of well-established clinical (Δ 6MWT), humoral (Δ BNP), and echocardiographic indexes (Δ LVEF, Δ LVESv) of CRTd response in the study cohorts, we found significantly different values of Δ 6MWT, Δ LVEF, Δ BNP, and Δ LVESv (*p* < 0.05, Figure 1).

Regarding the study outcomes, ED-CRTd patients were less likely to be CRTd responders, and had a higher number of hospitalizations for HF and cardiac deaths at follow-up compared with NED-CRTd (*p* < 0.05, Table 2).

In the Cox regression analysis, we found that hypertension (HR 0.818, CI 95% [0.669–0.999]), higher ET-1 (HR 0.859, CI 95% [0.839–0.979]), lymphocytes (HR 0.820, CI 95% [0.758–0.987]), LVEF (HR 0.876, CI 95% [0.760–0.992]), and ED (HR 0.751, CI 95% [0.624–0.905]) inversely predicted a lower likelihood of CRTd response, whereas higher miR-130a-5p (HR 1.490, CI 95% [1.014–2.188]) directly predicted CRTd response (Table 3).

Conversely, hypertension (HR 1.818, CI 95% [1.720–2.907]), higher BNP (HR 1.210, CI 95% [1.000–1.401]), and ED (HR 1.905, CI 95% [1.238–2.241]) predicted the rate of HF hospitalizations, whereas higher miR-130a-5p (HR 0.332, CI 95% [0.347–0.804]) and use of angiotensin receptor-neprilysin inhibitor (ARNI; HR 0.319, CI 95% [0.310–0.572]) inversely predicted lower rates of HF hospitalizations (Table 3).

The Cox curves showed the cumulative risk of being CRTd responders and requiring HF hospitalizations in NED-CRTd vs. ED-CRTd patients (Figure 2).

## 3. Discussion

In the present study, we diagnosed ED in CRTd patients according to the cut-off value of FMD ≤ 7.1% via echo color Doppler measurements of the brachial artery [1,2]. Compared with NED-CRTd patients, ED-CRTd patients had higher values of serum ET-1 at baseline, and displayed over-inflammation, higher serum ET-1, and lower serum values of miR-130a-5p at follow-up (1 year after CRTd implant). Furthermore, ED-CRTd patients had a worse clinical prognosis, with a lower rate of CRTd responders and higher rates of HF hospitalizations and cardiac deaths at 1-year follow-up. Intriguingly, higher serum expression of miR-130a-5p could predict the rate of CRTd responders and inversely predict HF hospitalizations; this was also found for ARNI therapy. Moreover, hypertension, serum ET-1, LVEF, and the diagnosis of ED were negative predictors of the rate of CRTd responders. Finally, hypertension, serum BNP values, and ED increased the risk of HF hospitalizations at 1 year of follow-up.

FMD values are associated with the presence of cardiovascular risk factors, structural arterial disease, cardiovascular outcomes, and changes in response to inflammatory stress [9]. Previous reports have suggested that CRTd was able to increase FMD and to reduce ET-1 expression, leading to better clinical outcomes [1,9]. Similarly, in our study, CRTd patients with ED (FMD ≤ 7.1) exhibited a reduction of ~25% of the probability of CRTd response and a 1.9-fold increased risk of HF hospitalization at 1-year follow-up. Conversely, higher values of serum ET-1 reduced the probability of CRTd response by ~14%. As mentioned above, ED and serum values of ET-1 may be modulated by favorable effects of CRTd. In this setting, CRTd may regulate FMD and ameliorate ED in CRTd patients [1,2], leading to a reduction in ET-1 and inflammatory markers [1,10]. 

Inflammation (e.g., a higher number of lymphocytes) was associated with an 18% lower probability of a CRTd response. Inflammation is a known risk factor that can affect the entity of a CRTd response and trigger cardiac injury, eventually worsening cardiovascular function [8,11,12,13,14,15,16,17]. Myocardial function is commonly assessed by LVEF values in CRTd patients, and is influenced by cardiac volumetry and contractile status [10,12,13,14,15,16,17,18]. Accordingly, we found that the lowest values of LVEF reduced the rate of CRTd responders at 1 year. A reduction of LVEF is frequently observed in CRTd patients with advanced cardiac remodeling and severe deficiency of cardiac pump function [10,12,13,14,15,16,17,18]. On the other hand, increased LVEF was detected among CRTd responders, which is linked with an improvement of cardiac pump function, hemodynamics, and clinical status, resulting in better clinical outcomes [10,12,13,14,15,16,17,18]. 

We also reported that high BNP values independently predicted the rate of HF hospitalizations. BNP is an established diagnostic marker of cardiac pump dysfunction and a strong prognostic predictor of patients at different stages of HF [9,10,12,13,14,15,16,17,18]. It is released in conditions of adaptive cardiac remodeling and in response to augmented inflammatory stress, myocardial tension, and increased intravascular volume [9,10,12,13,14,15,16,17,18]. 

Our study evidenced that higher values of miR-130a-5p led to a 1.5-fold increase in the rate of CRTd responders and decreased the risk of HF hospitalizations by 67%. As previously reported in an animal model of HF, the upregulation of miR-130a-5p significantly reduces inflammatory burden and ET-1 expression [7]. MiR-130a-5p plays a relevant role in chronic HF and in cardiac reparative mechanisms, acting on endothelial function and on the inflammatory burden [7,19]. Indeed, miR-130a-5p upregulation improved endothelial function via ET-1 downregulation in HF rats [7]. Similarly, we found significantly augmented levels of miR-130a-5p in NED-CRTd patients compared with ED-CRTd patients, accompanied by increased LVEF and reduced inflammatory burden and serum ET-1. On the other hand, at follow-up, ED-CRTd patients had over-inflammation, higher ET-1, and lower serum values of miR-130a-5p. Based on these findings, we suggest that the endothelium-dependent ameliorative mechanisms of CRTd may depend on the upregulation of serum miR-130a-5p in CRTd patients. Hence, miR-130a-5p may be seen as a favorable regulator of cardiac pump and endothelial functions in CRTd patients. Thus, miR-130a-5p may counter-regulate the multiple pathways implicated in cardiac remodeling, inflammation, and endothelial function [17,19,20].

Hypertension, a known risk factor for over-inflammation and a leading cause of ED, could worsen cardiovascular clinical outcomes [3,17]. In our study, hypertension reduced the rate of CRTd responders by ~18% and increased the risk of HF hospitalizations 1.8-fold. Therefore, worse clinical outcomes may be attributable to the negative effects exerted by hypertension on the vascular endothelium (evidenced by a reduced FMD). In this setting, we suggest that drugs modulating the ED, and showing cardiac anti-remodeling properties such as the ARNI, may revert this vicious circle. ARNI treatment has been shown to effectively reduce HF hospitalizations by approximately 68% [19]. ARNI show beneficial effects on the cardiac pump function and hemodynamics of HF patients, with an improved NYHA class, lowering of NT-proBNP values, and reduction of HF hospitalizations and death events [19]. According to a recent study, ARNI are also used in CRTd non-responders, who are patients with adverse cardiac remodeling and worse clinical outcomes [21]; in this group of CRTd patients, ARNI treatment could work as a modulator of miRs expression, with anti-remodeling cardiac effects, eventually leading to better clinical outcomes [21].

The current study is not exempt from limitations. The small sample size and limited duration of follow-up do not allow to drive definitive conclusions on the effects of CRTd on miR-130a-5p expression, modulation of ED, and clinical outcomes. To mechanistically imply miR-130a-5p as a central effector of complex pathological alterations of ED, further dedicated studies using animal models of chronic HF and ex-vivo models in isolated cardiomyocytes are also necessary. Such models would be extremely useful in testing the effects of mimics vs. antagomirs of miR-130a-5p on inflammatory, oxidative, and ED pathways at cellular and molecular levels via modifications of miR expression. This strategy may be useful in shaping and evaluating specific treatments with mimics and/or inhibitors of miR-130a-5p on ED. Nevertheless, further studies are warranted to address this hypothesis and identify novel molecular markers of reversed remodeling after CRTd in non-responders. Notably, in the current study, we did not fully analyze all the biomarkers identified as predictors of CRT response [22,23]. Thus, this aspect could limit the application of our findings in predicting the response to CRT. Finally, we did not enroll patients treated with CRT pacemakers (CRTp). On the other hand, we implanted CRTd because it was associated with lower mortality compared with CRT-*p* in HFrEF patients with severely reduced EF [24].

## 4. Materials and Methods

### 4.1. Study Design

We designed a prospective, observational, multicenter study, conducted between January 2017 and January 2021, with a follow-up duration of 1 year. We screened a population of consecutive HF patients with reduced left ventricle ejection fraction (LVEF), left bundle branch block, and indication to receive a CRTd [8], according to specific criteria.

**Inclusion criteria**: at least 18 years of age, with a clinical history of stable chronic HF; New York Heart Association (NYHA) functional class II or III; left bundle branch block; LVEF ≤ 35%; stable sinus rhythm; and indication to receive a CRTd according to the diagnostic criteria [8].

**Exclusion criteria:** age <18 or >75 years; ejection fraction ≥35%; patients in NYHA class IV; hyperkalemia; systolic hypotension (systolic blood pressure < 90 mmHg); estimated glomerular filtration rate (eGFR) < 30 mL per minute per 1.73 m^2^ of body surface area; evidence of atrial fibrillation (limitations in obtaining an accurate measure of FMD); patients with unstable HF, and those treated with an intravenous inotropic agent within 30 days before implantation (affects endothelial function); absence of written informed consent; and any condition that would make survival for 1 year unlikely.

We reported the effects of CRT-d in terms of clinical outcomes, CRT responder rate, and clinical events such as deaths, cardiac deaths, and hospitalizations for HF worsening. We then divided our population into two groups at baseline, according to FMD values obtained via echo color Doppler measurements of the brachial artery, with a cut-off at 7.1%, as previously reported [8,11]. The two groups were CRTd patients with ED (ED-CRTd; patients with FMD ≤ 7.1%) and CRTd patients without ED (NED-CRTd; patients with a FMD > 7.1%). The study was conducted according to the Declaration of Helsinki. The Ethics Committees of all participating institutions approved the protocol (number 247). All patients were informed about the nature of the study and gave signed informed consent to participate.

### 4.2. Anthropometric and Echocardiographic Evaluations and CRTd Implant 

In this investigation of ED-CRTd vs. NED-CRTd patients, we performed a physical examination, evaluated vital signs, and reviewed adverse events. A trans-thoracic two-dimensional echocardiogram with M-mode recordings, conventional Doppler, and pulsed-wave tissue Doppler imaging (TDI) measurements was performed at baseline and at the 12-month follow-up using Philips iE33 echocardiography (Eindhoven, The Netherlands). The left ventricle end-diastolic diameter (LVEDD), end-diastolic volume (LVEDV), end-systolic diameter (LVESD), and end-systolic volume (LVESV) were measured, and the LVEF was calculated with the Simpson method [12,13]. The grading of mitral regurgitation was classified as low (+), moderate (++), moderate-severe (+++), and severe (++++), [12,13]. Two trained physicians, blinded to the study protocol, performed the echocardiography and analyzed all echocardiographic data. Experienced electrophysiologists performed the CRTd implants. The final position of the CRTd leads was confirmed by catheter interrogation and cine fluoroscopy view [14,15]. 

### 4.3. Laboratory Analysis 

After an overnight fast for all patients, we evaluated plasma glucose, serum lipids, B-type natriuretic peptide (BNP), and N terminal pro-BNP (NT-proBNP) by enzymatic assays. We evaluated serum levels of pro-inflammatory cytokines (TNF-α, IL-1, and IL-6), systemic inflammatory markers (C-reactive protein, CRP), and leucocyte and neutrophil counts at baseline and 12-month follow-up [16,17]. For the determination of TNF-α, IL-1, IL-6, and CRP, commercially available enzyme-linked immunosorbent assay (ELISA) kits were used, according to the manufacturers’ protocols (TNF-α: TNF-alpha Human ELISA Kit KHC3011, ThermoFisher Scientific (Waltham, MA, USA); IL-1: Human IL-1 α ELISA Kit RAB0269, Sigma-Aldrich; IL-6: Human IL-6 Quantikine ELISA Kit D6050, R&D Systems; CRP: CRP Human ELISA kit KHA0031, ThermoFisher Scientific). An ice-cooled blood collection system was used to collect blood samples, which were immediately centrifuged for 10 min at 2.500 rpm at 4 °C. Supernatants containing serum samples were isolated and stored at −80 °C, before proceeding with ELISAs. We measured the plasma levels of ET-1 (pmol/L) using a chemiluminescence sandwich immunoassay (CT-proET-1 LIA, B.R.A.H.M.S GmbH, Hennigsdorf/Berlin, Germany) [19]. We referred to the analytical detection limit for plasma CT-proET-1 of 0.4 pmol/L, with an inter-laboratory variability coefficient of <10% for values > 10 pmol/L; the stability of CT-proET-1 in plasma at room temperature is at least 4 h [10]. 

### 4.4. RNA Serum Extraction and miR-130a-5p Analysis in CRTd Patients

To quantify miR-130a-5p expression, we used the miRNeasy Serum/PlasmaMini kit (217184, Qiagen) [17]; miR-130a-5p was assayed from blood samples collected at baseline and at 12 months in ED-CRTd and NED-CRTd patients. We selected miR-130a-5p as it is involved in the pathobiology of HF and has been linked to ED [17,18]. To monitor the efficiency of miR recovery and normalize miR expression, we spiked 5 nM Syn-cel-miRNA-39 miScript miRNA-Mimic before nucleic acid preparation [18]. A 5 µL aliquot of RNA was then reverse-transcribed using a miScript II RT kit (SABiosciences, Frederick, USA). Triplicate determinations of hsa-miR-130a-5p (MIMAT0004593MIMAT000025) were performed through CFX96 Real-Time System C1000 Touch Thermal Cycler (BioRad Laboratories, Inc., Hercules, California, USA) by using the miScript SYBR Green PCR kit (218073, Qiagen) and specific miScript primer assays (MS00008603, Qiagen) [4,7,18]. RT-qPCR data were analyzed using the 2-ΔΔCT method, where cycle threshold values were determined by CFX ManagerTM Software (BioRad Laboratories, Inc., Hercules, California, USA) [7,18,25]. We chose the sequences of the forward and reverse primers according to a previous study [26]. Then, we analyzed miR-130a-5p with the web-based software package for the miRNAPCR array system, as previously reported [26]. Furthermore, according to a previous study [13], we performed an miR target prediction with miRbase [26], and we found endothelial signaling as a predicted target of hsa-miR-130a-5p by TargetScanVert and miRDB.

### 4.5. Echo Doppler Measurements of the Brachial Artery and Assessment of Endothelial Function

Two experienced physicians evaluated the endothelial function via ultrasound examination of the brachial artery above the *antecubital fossa*, with a blood pressure cuff placed on the forearm distal to the 10 MHz linear-array ultrasound transducer probe, as recommended by international guidelines for the ultrasound assessment of endothelial-dependent flow-mediated vasodilation of the brachial artery [11]. Brachial artery reactivity was evaluated in the morning in a fasting, resting, and supine state in patients in a clinostatic position for 20 min at a room temperature of 22 °C [9]. We measured the peak of brachial artery blood flow velocity by the pulsed-wave Doppler signal at rest, at baseline, and during reactive hyperemia. The artery internal diameter, from the anterior to posterior wall at the interface between media and intima or to the hardest echo (‘‘M’’ line) at end-diastole, was also measured. We measured endothelium-dependent FMD as the maximum increase in the internal diameter of the brachial artery during reactive hyperemia, evoked by the release of an occlusion cuff inflated to 200 mm Hg for 5 min on the upper arm distal to the measurement site [1,2,9]. After cuff deflation, we measured the maximal peak flow rate at 15 and 30 s. We continuously recorded the changes in brachial artery diameter for 6 min [1,2,9]. We reported inter- and intra-observer variabilities in diameter measurements to be ≤3%.

### 4.6. Study Endpoints

Our study endpoints included clinical outcomes, serum inflammatory markers, and ET-1 and miR-130a-5p expression. Clinical outcomes were evaluated as primary study endpoints, and included the percentage of CRTd responders, hospitalizations for HF worsening, cardiac deaths, and all-cause deaths at 12-month follow-up. As secondary study endpoints, we evaluated the expression of serum inflammatory markers, ET-1 and miR-130a-5p, and their differences at 12 months of follow-up. 

All patients were instructed to regularly assess body weight, the occurrence of dyspnea, and any clinical symptoms, then report them at the clinical visit. Clinical evaluations included a physical examination, check of vital signs, and review of adverse events. A fasting blood test (at least 12 h from last meal) was performed for biochemical peripheral blood assay evaluation. 

According to international guidelines for HF and CRTd, we defined CRTd responders as HF patients fulfilling the following diagnostic criteria: evidence of left ventricular reverse remodeling (reduction in left ventricular end-systolic volume (LVESV) of ≥10% at cardiac echography), a significant change in functional HF class (improvement in the six min walk test and Minnesota Living with HF scale), and no HF hospitalizations at six months after CRTd implantation [8,11,12,13,14].

### 4.7. Statistical Analyses

Following verification of normality (Shapiro–Wilk’s test) and equal variance (Bartlett’s test), continuous variables were expressed as means and standard deviations and tested by a two-tailed Student *t*-test or Mann–Whitney test, whereas categorical variables were compared by chi-square or Fisher exact tests, as appropriate. Survival analysis was performed with the use of the Kaplan–Meier method. Predictors of the composite study endpoints A and B were evaluated using Cox regression models adjusted for potential confounders. A univariate analysis was conducted to examine the association between single main clinical factors, echocardiographic characteristics, serum ET-1, serum miR-130a-5p, and 12-month study outcomes. All variables with a *p*-value of less than 0.2 in the univariate analysis were subsequently entered into a multivariate model. In the multivariate model, a *p*-value of less than 0.05 was considered statistically significant. For all independent predictors, 95% confidence intervals were calculated. Statistical significance was established at a *p*-value < 0.05 for all the other analyses. The statistical analyses were performed using the SPSS software package (SPSS Inc., Armonk, NY, USA).

## 5. Conclusions

Serum values of miR-130a-5p, on top of established ultrasound indexes [20], could be useful in evaluating endothelial function in CRTd patients. Specifically, low values of miR-130a-5p could help in identifying patients at different stages of ED, and those with low predicted CRTd responses and worse prognoses. Taken together, our data identify miR-130a-5p as a novel and reliable diagnostic and therapeutic biomarker of ED in CRTd patients.

## 6. Clinical Perspectives

The clinical relevance of the current research relies on showing the importance of ED to predict CRTd outcomes, via the evaluation of FMD added to serum markers as miR-130 and ET-1. 

This aspect could imply, in clinical practice, the identification of new serum biomarkers of ED, namely miR-130 and ET-1, to predict poor prognoses in CRTd patients.

In the future, the evaluation of these markers, and the use of specific therapies to modulate their serum expression, could be applied in clinical setting to improve the outcomes in CRTd patients.

## Figures and Tables

**Figure 1 ijms-24-01510-f001:**
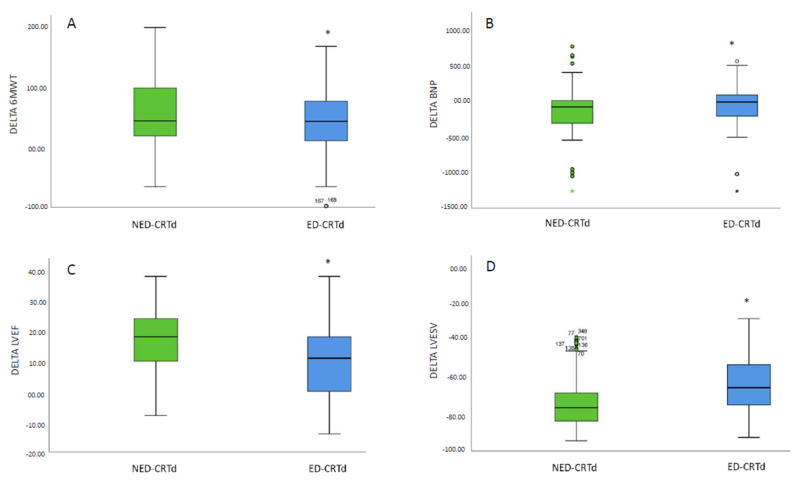
Comparison of differences (delta, Δ) at follow-up to baseline values of 6 min walking test (6MWT), B-type natriuretic peptide (BNP), left ventricle ejection fraction (LVEF), and left ventricle end systolic volume (LVESV) between CRTd patients with or without endothelial dysfunction (ED). We evaluated clinical (Δ 6MWT, **A**), humoral (Δ BNP, **B**), and echocardiographic indexes (Δ LVEF, **C** and Δ LVESv, **D**) of CRTd response in NED-CRTd (green) and ED-CRTd (blue) patients. CRTd: cardiac resynchronization therapy; ED-CRTd: CRTd patients with endothelial dysfunction; NED-CRTd: CRTd patients without endothelial dysfunction; 6MWT: six minutes walking test; BNP: B-type natriuretic peptide; LVEF: left ventricle ejection fraction; LVESv: left ventricle end-systolic volume. Boxes represent quartiles, whiskers represent 5–95 percentiles (outliers are shown as well). *: statistically significant, with *p* < 0.05.

**Figure 2 ijms-24-01510-f002:**
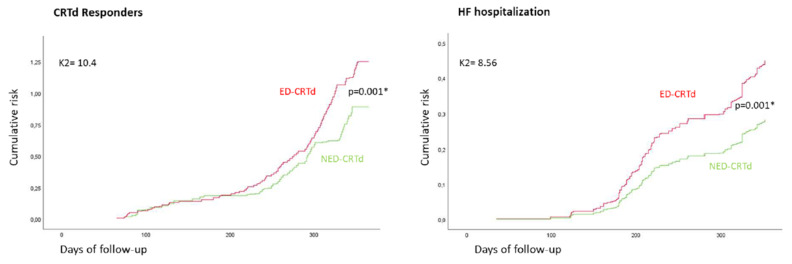
The Cox curves show the cumulative risk of study outcomes, being CRTd responders (**left part of figure**) and requiring HF hospitalizations (**right part of figure**) at 1-year follow-up in NED-CRTd (green) vs. ED-CRTd (red) patients. CRTd: cardiac resynchronization therapy; ED-CRTd: CRTd patients with endothelial dysfunction; NED-CRTd: CRTd patients without endothelial dysfunction. *: statistically significant, with *p* < 0.05.

**Table 1 ijms-24-01510-t001:** Clinical characteristics of study population at baseline and at end of follow-up for ED-CRTd vs. NED-CRTd. ACE: angiotensin-converting enzyme; ARS: angiotensin receptor; A.U.: arbitrary unit; BMI: body mass index; BNP: B-type natriuretic peptide; COPD: chronic obstructive pulmonary disease; CRP: C-reactive protein; ED: endothelial dysfunction; IL6: interleukin 6; miR-130: microRNA-130; NYHA: New York Heart Association; LVEF: left ventricle ejection fraction; LVEDd: left ventricle end diastolic diameter; LVEDv: left ventricle end diastolic volume; LVESd: left ventricle end systolic diameter; LVESv: left ventricle end systolic volume; mitral insufficiency +: low-grade; ++: moderate; +++: more than moderate; NOAC: new oral anticoagulant; SGLT2-I: sodium–glucose transporter 2 inhibitors; NED: non-endothelial dysfunction; 6MWT: six minutes walking test; TNFa: tumor necrosis factor alpha; *: *p* < 0.05 (statistical significant).

	BASELINE			1-YEAR FOLLOW-UP		
PARAMETERS	ED-CRTd (FMD ≤ 7.1, n 590)	NED-CRTd (FMD ≥ 7.1, n 277)	*p* Value	ED-CRTd (FMD ≤ 7.1, n 326)	NED-CRTd (FMD ≥ 7.1, n 541)	*p* Value
Age, years	70.7 ± 6.2	71.1 ± 5.8	0.568	71.7 ± 6.6	71.8 ± 6.3	0.749
Male, n (%)	429 (72.7)	199 (71.8)	0.060	233 (71.5)	393 (72.6)	0.754
Smokers, n (%)	295 (49.5)	128 (46.2)	0.102	179 (54.9)	283 (52.3)	0.483
Hypertension, n (%)	417 (70.7)	187 (67.5)	0.344	232 (71.2)	369 (68.2)	0.363
Dyslipidemia, n (%)	257 (43.6)	110 (39.7)	0.302	115 (35.3)	184 (34.0)	0.713
Diabetes mellitus, n (%)	249 (42.2)	109 (39.3)	0.376	158 (48.5)	229 (42.3)	0.090
BMI > 30 kg/m^2^ (%)	45 (7.6)	19 (6.9)	0.781	32 (9.8)	34 (6.3)	0.064
Ischemic heart failure (%)	409 (69.3)	186 (67.1)	0.531	247 (75.8)	383 (70.8)	0.116
NYHA class, n (%):			0.488			0.001 *
I NYHA class	/	/		7 (2.1)	35 (6.5)	
II NYHA class	130 (22.0)	67 (24.2)		69 (21.2)	266 (49.2)	
III NYHA class	460 (78)	210 (75.8)		219 (67.2)	223 (41.2)	
IV NYHA class	/	/		31 (9.5)	17 (3.1)	
QRS duration (ms)	137.8 ±9.2	138.0 ± 9.5	0.160	127.2 ±6.2	120.6 ± 9.6	0.001 *
6MWT	209.56 ± 44.15	208.16 ± 44.53	0.118	218.17 ± 44.15	247.17 ± 44.52	0.018 *
BNP (pg/mL)	390.95 ± 29.34	402.33 ± 23.01	0.570	297.43 ± 16.22	266.25 ± 10.8	0.042 *
Endothelin-1, pmol/L	6.49 ± 0.18	5.63 ± 0.25	0.007*	5.41 ± 0.24	4.57 ± 0.17	0.003 *
miR-130a-5p, A.U.	0.28 ± 0.014	0.27 ± 0.025	0.688	0.41 ± 0.034	0.51 ± 0.029	0.037 *
**Inflammatory biomarkers**						
Lymphocytes	7.13 ± 1.36	7.46 ± 1.52	0.438	7.93± 1.83	6.93± 1.12	0.001 *
Neutrophiles	5.83 ± 1.06	5.70 ± 1.23	0.071	5.73 ± 0.92	5.24 ± 1.20	0.001 *
CRP (pg/l) x 10	9.26 ± 0. 41	8.96 ± 0.51	0.676	9.86 ± 0. 48	6.59 ± 0.38	0.001 *
IL6 (pg/mL)	6.48 ± 0.02	6.52 ± 0.03	0.462	6.30 ± 0.06	6.10 ± 0.06	0.036 *
TNFα (pg/mL) x 10	6.43 ± 0.02	6.47 ± 0.02	0.144	6.38 ± 0.02	6.16 ± 0.02	0.001 *
**Echocardiographic parameters**						
LVEF (%)	26.8 ± 5.4	26.3 ± 4.9	0.126	36.7 ± 6.9	42.6 ± 4.5	0.001 *
LVEDd (mm)	68.2 ± 4.1	69.1 ± 3.7	0.968	71.7 ± 5.8	68.1 ± 3.9	0.001 *
LVESd (mm)	42.6 ± 5.3	43.2 ± 6.0	0.786	41.5 ± 3.8	38.6 ± 4.8	0.001 *
LVEDv (ml)	224.8 ± 22.1	227.1 ± 24.3	0.335	228.4 ± 19.7	219.2 ± 14.1	0.001 *
LVESv (ml)	140.2 ± 22.5	139.1 ± 23.8	0.328	137.25 ± 16.6	124.8 ± 17.2	0.001 *
Mitral insufficiency			0.384			0.050 *
+ (%)	272 (46.1)	135 (48.7)		130 (39.9)	265 (49.0)	
++ (%)	230 (38.9)	108 (39.0)		131 (40.2)	219 (40.5)	
+++ (%)	88 (14.9)	34 (12.3)		62 (19.9)	57 (10.5)	
**Medications**						
Beta blockers, n (%):	405 (68.6)	188 (67.9)	0.876	237 (72.7)	380 (70.2)	0.487
Carvedilol	291 (71.9)	137 (72.9)		174 (73.4)	281 (73.9)	
Bisoprolol	114 (28.1)	51 (27.1)		63 (26.6)	99 (26.1)	
Calcium antagonist, n (%)	23 (3.9)	9 (3.2)	0.703	13 (4.0)	22 (4.1)	0.159
Amiodarone, n (%)	117 (19.8)	60 (21.7)	0.588	82 (25.1)	108 (20.0)	0.076
ACE inhibitors, n (%)	148 (25.1)	68 (24.5)	0.867	88 (27.0)	143 (26.4)	0.575
ARS blockers, n (%)	167 (28.3)	84 (30.3)	0.574	95 (29.1)	165 (30.5)	0.597
Sacubitril/valsartan, n (%)	188 (31.9)	93 (33.6)	0.641	132 (40.5)	177 (32.7)	0.023 *
Aspirin, n (%)	224 (38.0)	111 (40.1)	0.601	134 (41.1)	211 (39.0)	0.424
Warfarin, n (%)	199 (33.7)	102 (36.8)	0.646	124 (38.0)	185 (34.2)	0.272
NOAC, n (%)	117 (19.8)	55 (19.8)	0.928	69 (21.2)	112 (20.7)	0.421
Ticlopidine, n (%)	10 (1.7)	5 (1.8)	0.826	8 (2.4)	11 (2.0)	0.496
Ivabradine, n (%)	183 (31.0)	78 (28.2)	0.473	(30.9)	(28)	0.822
Digoxin, n (%)	178 (30.2)	91 (32.8)	0.387	(30.2)	(32.8)	0.766
Diuretics, n (%):						
Loop diuretics	526 (89.1)	246 (88.8)	0.602	303 (92.9)	472 (87.2)	0.041 *
Tiazides	70 (11.9)	30 (10.8)	0.737	41 (12.6)	60 (11.1)	0.516
Aldosterone Blockers	384 (65.1)	185 (66.8)	0.433	228 (66.9)	367(67.8)	0.597
Statins, n (%)	416 (70.5)	197 (71.1)	0.810	(72.1)	(72.4)	0.875
SGLT2-I, n (%)	124 (21.0)	61 (22.0)	0.723	98 (30.1)	124 (22.9)	0.020 *

**Table 2 ijms-24-01510-t002:** Clinical outcomes at 12-month follow-up for ED-CRTd vs. NED-CRTd patients. CRTd: cardiac resynchronization with a defibrillator; ED: endothelial dysfunction; NED: non-endothelial dysfunction; n: number. *: statistically significant, with *p* < 0.05.

Study Outcomes at A 1 Year of Follow-Up	Overall Population	ED-CRTd	NED-CRTd	*p* Value
CRTd responder rate, n (%)	569 (65.6)	189 (58)	380 (70.2)	0.001 *
Hospitalization for heart failure, n (%)	269 (31.0)	115 (35.3)	154 (28.5)	0.041 *
Cardiac deaths, n (%)	51 (5.8)	30 (9.2)	21 (3.9)	0.002 *
All-cause deaths, n (%)	52 (5.9)	25 (7.7)	27 (5.0)	0.139

**Table 3 ijms-24-01510-t003:** Multivariable Cox regression analysis for assessing independent predictors of CRTd responders (**A**) and HF hospitalizations (**B**). T2DM: type 2 diabetes mellitus; 6MWT: six minutes walking test; BNP: B-type natriuretic peptide; miR-130a-5p: microRNA-130a-5p; CRP: C-reactive protein; LVEF: left ventricle ejection fraction; ARNI: angiotensin receptor-neprilysin inhibitor; NYHA 3: New York Heart Association class 3; BB: beta blocker; ED: endothelial dysfunction. *: statistically significant, with *p* < 0.05.

(A)		UNIVARIATE ANALYSIS			MULTIVARIATE ANALYSIS	
		CRTd Responders			CRTd Responders	
Risk Factors	HR	95% CI	*p* Value	HR	95% CI	*p* Value
Age	0.998	0.984–1.012	0.737			
Hypertension	1.380	1.145–1.664	0.001 *	0.818	0.669–0.999	0.049 *
Obesity	0.849	0.620–1.163	0.309			
T2DM	0.682	0.577–1.807	0.185			
6MWT	0.989	0.907–1.001	0.165			
BNP	1.012	0.999–1.100	0.120			
CRP	1.012	1.004–1.020	0.004 *	1.007	0.998–1.015	0.133
Lymphocytes	0.922	0.863–0.985	0.016 *	0.820	0.758–0.987	0.009 *
miR-130a-5p	1.826	1.106–2.306	0.036 *	1.490	1.014–2.188	0.042 *
Endothelin-1	0.981	0.743–0.995	0.043 *	0.859	0.839–0.979	0.001 *
LVEF	0.976	0.961–0.991	0.002 *	0.876	0.760–0.992	0.004 *
ARNI	1.034	0.866–1.235	0.710			
NYHA 3	0.812	0.685–0.963	0.017*	0.844	0.672–1.059	0.143
BB	0.986	0.826–1.178	0.879			
ED	0.362	0.153–0.609	0.001*	0.751	0.624–0.905	0.003*
**(B)**		**UNIVARIATE** **ANALYSIS**			**MULTIVARIATE ANALYSIS**	
		**HF Hospitalizations**			**HF Hospitalizations**	
**Risk Factors**	**HR**	**95% CI**	***p* value**	**HR**	**95% CI**	***p* value**
Age	0.989	0.969–1.019	0.900			
Hypertension	1.738	1.575–1.947	0.017 *	1.818	1.720–2.907	0.001 *
Obesity	0.971	0.622–1.516	0.898			
T2DM	1.010	1.000–1.101	0.001 *			
6MWT	0.998	0.996–1.001	0.190			
BNP	1.011	1.000–1.102	0.001 *	1.210	1.000–1.401	0.047 *
CRP	0.983	0.969–0.997	0.018 *	1.007	0.978–1.008	0.345
Lymphocytes	1.083	0.986–1.190	0.097	0.987	1.022–1.266	0.180
miR-130a-5p	0.566	0.384–0.835	0.004 *	0.332	0.347–0.804	0.003 *
Endothelin-1	1.006	0.979–1.034	0.668			
LVEF	1.026	1.003–1.050	0.029 *	0.992	0.986–1.038	0.394
ARNI	0.160	0.086–0.563	0.001 *	0.319	0.310–0.572	0.001 *
NYHA 3	1.071	0.843–1.360	0.576			
BB	0.828	0.645–1.063	0.139			
ED	1.301	1.232–1.390	0.001 *	1.905	1.238–2.241	0.001 *

## Data Availability

Data and materials are available on request.

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
