# Peer review of "Endothelial Dysfunction Drives CRTd Outcome at 1-Year Follow-Up: A Novel Role as Biomarker for miR-130a-5p"

_ijms, 2023, doi:10.3390/ijms24021510_

Round 1

Reviewer 1 Report

In this manuscript the authors conducted an observational multicenter prospective study to investigate the potential role of the miR-130a-5p as a novel biomarker of Endothelial Dysfunction through the pathway mediated by the Endothelin-1 (ET-1). The study is well designed, the experimental procedure is correctly conducted and the statistical analysis of the data is appropriated.  The role of the miR-130a-5p in endothelial dysfunction is amply reported in the literature, however with this study the authors have made some innovations such as the utilization of this microRNA as biomarker of treatment response.

Therefore, I think that the manuscript should be accepted for publication but I would like some clarifications/changes from the authors which could improve the manuscript.

Q1: The authors in the introduction indicated “Emerging evidence indicates that ET-1 is directly targeted by microRNA-130 (miR-130a-5p), [9]”. Are these evidences also supported by the main microRNA database, miRbase? I believe that this information would provide some useful information on the “seed sequence” of the microRNA that interact with the target, EDN1 that encodes for the  ET-1 protein.  

Q2:  In the Table 1, the distribution of male subject between the two groups at the baseline is different in the groups respect to 1-year follow up. This dissimilarity could not influence the results obtained? I would like the authors to clarify this point.

Q3: In the description of the sample size it would be better to write “n =” and not “n”

Author Response

In this manuscript the authors conducted an observational multicenter prospective study to investigate the potential role of the miR-130a-5p as a novel biomarker of Endothelial Dysfunction through the pathway mediated by the Endothelin-1 (ET-1). The study is well designed, the experimental procedure is correctly conducted and the statistical analysis of the data is appropriated.  The role of the miR-130a-5p in endothelial dysfunction is amply reported in the literature, however with this study the authors have made some innovations such as the utilization of this microRNA as biomarker of treatment response. Therefore, I think that the manuscript should be accepted for publication but I would like some clarifications/changes from the authors which could improve the manuscript.

We thank the reviewer for his/her positive comment.

Q1: The authors in the introduction indicated “Emerging evidence indicates that ET-1 is directly targeted by microRNA-130 (miR-130a-5p), [9]”. Are these evidences also supported by the main microRNA database, miRbase? I believe that this information would provide some useful information on the “seed sequence” of the microRNA that interact with the target, EDN1 that encodes for the ET-1 protein. 

We thank the reviewer for the helpful comment. According to your suggestion, we changed the previous reference reported [9] with this new one: “Bertero T, Cottrill K, Krauszman A, Lu Y, Annis S, Hale A, Bhat B, Waxman AB, Chau BN, Kuebler WM, Chan SY. The microRNA-130/301 family controls vasoconstriction in pulmonary hypertension. J Biol Chem. 2015 Jan 23;290(4):2069-85. doi: 10.1074/jbc.M114.617845”. Indeed, in this study (ref 11) the authors performed a microrna target prediction by using TargetScan 6.2 algorithm with Pathway enrichment analysis by using Reactome, Biocarta, NCBI PID, and KEGG databases, and the endothelin signaling emerged as target of miR-130/3012 family. However, according to the Reviewer’s suggestion, we checked this result by using miRbase and we found Endothelial signaling as a predicted target of hsa-miR-130a-5p by TargetScanVert and miRDB. Therefore, to better clarify this point, we described this informatic analysis in the Methods, section 4.4 as follows: “Furthermore, according to authors [11], we performed a miR target prediction with miR-base [21], and we found endothelial signaling as a predicted target of hsa-miR-130a-5p by TargetScanVert and miRDB”. You could see it at page 9, at the end (last lines) of section 4.4.

Thus, we also added this new reference: “Griffiths-Jones S, Grocock RJ, van Dongen S, Bateman A, Enright AJ. miRBase: microRNA sequences, targets and gene nomenclature. Nucleic Acids Res. 2006 Jan 1;34(Database issue):D140-4. doi: 10.1093/nar/gkj112”, to support this sentence.

Q2:  In the Table 1, the distribution of male subject between the two groups at the baseline is different in the groups respect to 1-year follow up. This dissimilarity could not influence the results obtained? I would like the authors to clarify this point.

At baseline, we had 429 (72.7%) vs. 199 (71.8%) male patients in the ED-CRTd vs. NED-CRTd cohorts (p 0.060). At follow-up end, we had 233 (71.5%) vs. 393 (72.6%) male patients in the ED-CRTd vs. NED-CRTd cohorts (p 0.754). Thus, as suggested we had a different number and rate of males in ED-CRTd vs. NED-CRTd patients. On the other hand, at baseline and at follow-up end this difference and dissimilarity of male patients did not reach the statistical significant between cohorts (at baseline, p 0.060); at follow-up end, p 0.754). Conversely, to analyze the effect of this dissimilarity on the study results, we added the male variable as risk factor in the Cox Univariate regression analysis for study outcomes (CRTd responders, and HF hospitalizations) at follow-up end. Thus, male variable did not influence the study outcomes: CRTd responders (HR 0.835, CI 95% [0.689-1.011], p 0.065), and HF hospitalizations (HR 2.306, CI 95% [0.324-6.431], p 0.404).  Therefore, we might speculate that the dissimilarity of the distribution of male subject between the two groups at the baseline respect to 1-year follow up did not influence study outcomes.

Q3: In the description of the sample size it would be better to write “n =” and not “n”.

As suggested, we corrected it.

Reviewer 2 Report

General Comments

The manuscript is well-written with broad referencing. The authors investigated “Endothelial dysfunction drives CRTd outcome at 1-year of follow-up: a novel role as biomarker for miR-130a-5p”. The study is interesting and adds to the existing body of knowledge. There are a few things that need clarification and revisions. All details and comments are listed below.

Details Comments

1.     Abstract, Line 7-8: ED-CRTd (n = 326), NED-CRTd (n = 541)

2.     Page 2, Introduction: Please briefly explain the mechanism ET-1 cause endothelial dysfunction. Please also explain how inflammation markers (TNF-a, IL-1, IL-6) increase ET-1 and endothelial dysfunction. 

3.     Page 2, Results, Line 4: p < 0.05

4.     Page 8, Materials and method, Study design: Please add human ethical voucher no./ ethical approval number

5.     Page 9, Laboratory analysis: Please briefly explain how the inflammatory markers (TNF-a, IL-1, IL-6, CRP) was measured. If the authors used an ELISA kit, please add details (brand and catalog number). Please explain how the blood was processed before the determination of those markers. 

6.     Page 9, RNA serum extraction: Please add the catalog number for the miRNeasy Mini kit from Qiagen.

7.     Page 9, RNA serum extraction: Please briefly explain the RT-qPCR process (please add primers; forward and reversed used with citations)

8.     Page 11, Ethics approval: Please add code/voucher for human ethical approval

Author Response

Reviewer 2#

The manuscript is well-written with broad referencing. The authors investigated “Endothelial dysfunction drives CRTd outcome at 1-year of follow-up: a novel role as biomarker for miR-130a-5p”. The study is interesting and adds to the existing body of knowledge. There are a few things that need clarification and revisions. All details and comments are listed below.

We thank the reviewer for his/her positive comment.

Details Comments

  1. Abstract, Line 7-8: ED-CRTd (n = 326), NED-CRTd (n = 541).

As suggested, we corrected it.

  1. Page 2, Introduction: Please briefly explain the mechanism ET-1 cause endothelial dysfunction. Please also explain how inflammation markers (TNF-a, IL-1, IL-6) increase ET-1 and endothelial dysfunction.

As suggested, we briefly explained in the Introduction the mechanism ET-1 cause endothelial dysfunction, and how inflammation markers (TNF-a, IL-1, IL-6) increase ET-1 and endothelial dysfunction. You could see it at page 2, lines 8-17.

  1. Page 2, Results, Line 4: p < 0.05.

As suggested, we corrected it.

  1. Page 8, Materials and method, Study design: Please add human ethical voucher no./ ethical approval number.

We added it as required.

  1. Page 9, Laboratory analysis: Please briefly explain how the inflammatory markers (TNF-a, IL-1, IL-6, CRP) was measured. If the authors used an ELISA kit, please add details (brand and catalog number). Please explain how the blood was processed before the determination of those markers.

According TO your suggestion, we measured the inflammatory markers (TNF-a, IL-1, IL-6, CRP) by commercially available assays. However, as required, we added all the brands and catalogue numbers, along with the description of blood processing, in section 4.3 (page 9, lines 12-20).

  1. Page 9, RNA serum extraction: Please add the catalog number for the miRNeasy Mini kit from Qiagen.

The required catalogue number has been added at the beginning of section 4.4, at first two lines of page 9.

  1. Page 9, RNA serum extraction: Please briefly explain the RT-qPCR process (please add primers; forward and reversed used with citations)

The primers for mir-130a-5p (MIMAT0004593) have been predesigned, synthetized and validated by Qiagen kit. Thus, as required we reported the Qiagen catalogue number (MS00008603 miscript Primer Assay) and all the missing information in section 4.4. We re-wrote this section. According to your suggestion, we added a new sentence and a new reference (ref 17: Liu HL, Bao HG, Zheng CL, Teng C, Bai MH. MiR-130a regulating the biological function of colon cancer by targeting inhibition of PTEN. Eur Rev Med Pharmacol Sci. 2020 Feb;24(4):1786-1793. doi: 10.26355/eurrev_202002_20356), to report the forward and reversed primers used. You could see the new sentence “We chose the sequence of the forward and reversed primers used according to authors [17]” in the text, section 4.4.

  1. Page 11, Ethics approval: Please add code/voucher for human ethical approval.

As required, we added it.

Reviewer 3 Report

Manuscript submitted by Sardu et al had focused on the application potential of a microRNA, miR-130A-5P in prediction of the endothelial dysfunction in heart failure patients after cardia resynchronization therapy. By designing a prospective multi-center study, authors of the study had proved the serrum miR-130a-5p served as an HF biomarker which was sufficient to identify different stages of ED and independently predict CRTd response. The study was well designed, the conclusion was draw based on sufficient numbers of samples, and the manuscript is well drafted.

1.     In the Abstract, line 4, why a “However,” is used here? Should “Therefore,” a better choice?

2.     In the Abstract, please also use mean±std for all values presented.

Author Response

Reviewer 3#

Manuscript submitted by Sardu et al had focused on the application potential of a microRNA, miR-130A-5P in prediction of the endothelial dysfunction in heart failure patients after cardia resynchronization therapy. By designing a prospective multi-center study, authors of the study had proved the serrum miR-130a-5p served as an HF biomarker which was sufficient to identify different stages of ED and independently predict CRTd response. The study was well designed, the conclusion was draw based on sufficient numbers of samples, and the manuscript is well drafted.

We thank the reviewer for the positive comment.

  1. In the Abstract, line 4, why a “However,” is used here? Should “Therefore,” a better choice?

According to your suggestion, we changed “However”, with “Therefore”.

  1. In the Abstract, please also use mean±std for all values presented.

As suggested, we added mean ± std for all values presented.

Round 2

Reviewer 1 Report

Thank you to authors to modify the article. I think that the manuscript is improved and suitable for the publication.